# Posttransplant Complications and Genetic Loci Involved in Telomere Maintenance in Heart Transplant Patients

**DOI:** 10.3390/genes13101855

**Published:** 2022-10-14

**Authors:** Dana Dlouha, Jevgenija Vymetalova, Sarka Novakova, Pavlina Huckova, Vera Lanska, Jaroslav Alois Hubacek

**Affiliations:** 1Center for Experimental Medicine, Institute for Clinical and Experimental Medicine, 140 21 Prague, Czech Republic; 2Cardio Center, Institute for Clinical and Experimental Medicine, 140 21 Prague, Czech Republic; 3Statistical Unit, Institute for Clinical and Experimental Medicine, 140 21 Prague, Czech Republic; 43rd Department of Internal Medicine, 1st Faculty of Medicine, Charles University, 140 21 Prague, Czech Republic

**Keywords:** telomere, heart transplantation, SNP, rejection, genetic risk score

## Abstract

Reaching critically short telomeres induces cellular senescence and ultimately cell death. Cellular senescence contributes to the loss of tissue function. We aimed to determine the association between variants within genes involved in telomere length maintenance, posttransplant events, and aortic telomere length in heart transplant patients. DNA was isolated from paired aortic samples of 383 heart recipients (age 50.7 ± 11.9 years) and corresponding donors (age 38.7 ± 12.0 years). Variants within the *TERC* (rs12696304), *TERF2IP* (rs3784929 and rs8053257), and *OBCF1* (rs4387287) genes were genotyped, and telomere length was measured using qPCR. We identified similar frequencies of genotypes in heart donors and recipients. Antibody-mediated rejection (AMR) was more common (*p* < 0.05) in carriers of at least one G allele within the *TERF2IP* locus (rs3784929). Chronic graft dysfunction (CGD) was associated with the *TERC* (rs12696304) GG donor genotype (*p* = 0.05). The genetic risk score did not determine posttransplant complication risk prediction. No associations between the analyzed polymorphisms and telomere length were detected in either donor or recipient DNA. In conclusion, possible associations between *donor TERF2IP* (rs3784929) and AMR and between *TERC* (rs12696304) and CGD were found. SNPs within the examined genes were not associated with telomere length in transplanted patients.

## 1. Introduction

Heart transplantation remains the only definitive therapy for advanced heart failure. Short-term survival after cardiac transplantation has improved significantly as a consequence of improved immunosuppressive therapy, monitoring, and surgical techniques. In contrast, long-term survival has a key barrier—the persistent prevalence of long-term complications such as malignancy and cardiac allograft vasculopathy (CAV). Patients, however, are also exposed to other traditional risk factors for native heart failure, such as hypertension, diabetes, coronary atherosclerosis, pericardial disease, and cardiac drug toxicity [1,2]. The most frequent posttransplant complications, such as rejection (either acute cellular—ACR or antibody-mediated—AMR), chronic graft dysfunction (CGD), and CAV, remain without a molecular-genetic explanation.

Telomeres (TLs) are tandem DNA repeat sequences localized at the ends of chromosomes that maintain genomic stability during cell division. Human telomeres range in size from 2 to 50 kilobases and consist of approximately 300–8000 hexanucleotide repeats. Telomeres shorten with each cell division because of ineffective replication of the 3′ end of the DNA chain. The “end replication problem” leads to the critical shortening of TL or instability of the telomeric binding proteins. This induces an arrest of cell division or replicative senescence [3], which may cause genomic instability. Achievement of critically short telomeres prompts cellular senescence and ultimately cell death [4,5,6].

Variants within the genes coding telomerase components that are important for telomere maintenance and integrity (*OBCF1*, *TERC*, and *TERT*) or protein inhibitors of DNA repair machinery were previously reported to influence leukocyte telomere length (LTL) [7,8,9,10,11,12].

Polymorphisms within the *TERC* gene (telomerase RNA component; OMIM ID 602322) [8] result in a reduction in telomerase activity leading to premature telomere shortening [13] and are associated with LTL [14,15]. The haplotype of three SNPs within *TERC* was reported to be associated with an increased risk of cardiovascular diseases (CVDs) and T2DM in European Caucasians [9], and other *TERC* variants have been associated with longevity in humans [7]. *TERC* rs12696304 was recently reported as a risk variant in the prognosis of mortality in a group of acute heart failure patients [16].

*TERF2IP* (TRF2-interacting telomeric protein; also known as *RAP1*; OMIM ID 605061) encodes a multifunctional protein located in the nucleus and cytoplasm. TERF2IP is a part of the shelterin complex that protects telomeres from degradation and prevents end-to-end fusion, atypical recombination, and premature senescence [17]. Previously, two SNPs (rs3784929; rs8053257) within the *TERF2IP* locus have been described to be associated with a greater risk of ischemic stroke in obese women [18,19].

*OBFC1* (oligonucleotide/oligosaccharide binding fold containing 1; OMIM ID 613128) is involved in the initiation of telomere replication. Several genome-wide association studies (GWAS) have associated variants within the *OBFC1* gene with LTL [20,21].

Our study focused on the analysis of variants within genes involved in telomere length maintenance (*TERC*, *TERF2IP*, and *OBCF1*) and their possible connection with posttransplant complications as well as with donor/recipient aortic telomere length (ArTL).

## 2. Materials and Methods

### 2.1. Subjects

Patients with chronic irreversible renal failure were contraindicated for orthotopic heart transplantation (OHT). Aortic biopsies of patients undergoing OHT were collected from January 2005 to December 2015 at the Institute for Clinical and Experimental Medicine in Prague as we described previously [22,23]. Briefly, aortic samples from 383 heart recipients (age 50.7 ± 11.9 years) and paired corresponding donors (age 38.7 ± 12.0 years) were obtained during OHT. Samples with low-quality DNA or repeated OHT were excluded from the analysis (total N = 9). The protocol of this study was carried out according to the principles of the Declaration of Helsinki [24]. All examined individuals provided their informed consent, which, together with the study protocol, was approved by the institution’s ethics committee.

Endomyocardial biopsy (EMB) was used for surveillance of cardiac allograft rejection and for the diagnosis of unexplained ventricular dysfunction. On the same day, transthoracic echocardiography was also performed. Acute cellular rejection (ACR) was defined according to the Revision of the 1990 Working Formulation for the Standardization of Nomenclature in the Diagnosis of Heart Rejection [25]. Antibody-mediated (humoral) rejection (AMR) was defined according to the recommendations of the pathology task force of the International Society of Heart and Lung Transplantation (ISHLT) [26]. Diagnostic coronary angiography was used for the examination of CAV.

### 2.2. DNA Analysis

Genomic DNA was isolated from a maximum of 100 mg of the aortic samples using the modified “salt out” method [27] after cell lysis by Proteinase K (ThermoScientific, Waltham, MA, USA).

The quantity and purity of isolated DNA were examined by standard spectrophotometry using a Nanodrop 2000 spectrophotometer (ThermoScientific, Waltham, MA, USA). DNA integrity was tested on a 0.7% agarose gel (Bio Rad Power Pac 300, Hercules, CA, USA) after visualization using ethidium bromide.

### 2.3. Measurement of Telomere Length

Relative telomere length was analyzed as described previously [28] with slight modifications [12,29]. Briefly, the relative telomere length was calculated as the ratio of telomere repeats to single-copy gene (SCG) copies (T/S ratio). The acidic ribosomal phosphoprotein PO (36B4) gene was used as the SCG. For each sample, the quantity of telomere repeats and the quantity of SCG copies were determined compared to a reference sample. Identical reference DNA was used in all runs to allow comparisons of the results in different runs. Samples were analyzed in triplicate using the Rotor-Gene 3000 (Corbett Research Ltd., Mortlake, Australia).

### 2.4. SNPs Analysis

Genetic variants within *TERC* (rs12696304, assay ID C_407063_ 10), *TERF2IP* (rs3784929, assay ID C_25800757_10; rs8053257, assay ID C_31423806_10) and *OBFC1* (rs4387287, assay ID C_2818531_10) were analyzed using TaqMan technology (quantitative PCR-based method) on an AB 7300 RT PCR instrument (ThermoScientific, Waltham, MA, USA). To ensure the accuracy of genotyping, one plate (containing 94 DNA samples) was measured twice within one week with 100% conformity.

### 2.5. Statistical Analysis

Data are presented as percentages for categorical variables and mean ± standard deviation (SD) for continuous variables unless otherwise indicated. The Hardy–Weinberg test was used to confirm independent segregation of the alleles of individual polymorphisms (www.tufts.edu/~mcourt01/Documents/Court%20lab%20-%20HW%20calculator.xls, accessed on 25 April 2018). The frequencies of individual genotypes between the two groups were compared with the chi-square test. The Shapiro–Wilk W test for normality of telomere length was used and Anova was used for hypothesis testing. Analyses were performed with JMP 10 statistical software. ArTL values were available in 94.5% of all samples [23].

To analyze the genetic predisposition for posttransplant complications, the unweighted genetic risk score (GRS) was created according to the simple cumulative presence of different alleles—2 points for risk (shorter telomere-associated) allele homozygotes, 1 point for heterozygotes, and 0 points for subjects without any risk allele. Cochran Armitage Trend tests were used for the comparison of GRS between groups of posttransplant complications. GRS was calculated for both recipient and donor genotypes. *p* values less than 0.05 were considered significant.

## 3. Results

### 3.1. Main Characteristics

The basic characteristics of the study group are described in Table 1. Allograft recipients were older than donors (*p* < 0.0001), and the sex ratio was similar in both groups. Dilated cardiomyopathy (N = 163) and coronary artery disease (N = 155) were diagnosed as dominant primary heart diseases. The left ventricular assist device (LVAD), as a bridge to transplantation, was implanted in 30% of patients. The median LVAD duration was 201 days (range 7 to 2509 days). The most frequent posttransplant complication was ACR (grade > 1R; N = 142), with a majority of patients (N = 101) diagnosed during the first 6 months after OHT. AMR grade ≥ 1 developed in 35 subjects, 16 patients were affected during the first 6 months, and 15 patients were diagnosed later than 1 year after OHT. Of the other complications, CAV and CGD were detected in 58 and 57 patients, respectively, after OHT. From the study cohort, 157 patients had already died. The median survival time (ST) was 4.7 years (interquartile range of 0.82 to 8.6 years).

### 3.2. Genotyping

The call rate for genotyping was 99–100% for all SNPs. Distributions of individual genotypes were within the Hardy–Weinberg equilibrium. The frequencies of genotypes did not differ between recipients and donors (Table 2). AMR was more common (*p* = 0.048) in carriers of at least one donor G (but not recipient) allele at the *TERF2IP* locus (rs3784929). Furthermore, the donor GG genotype within the *TERC* gene (rs12696304) was linked with CGD (*p* = 0.055). Other polymorphisms were not significantly associated with any of the main posttransplant complications (Table 3) in the investigated samples. Neither heart failure etiology nor survival time of patients in our study were related to genotyped SNPs. Furthermore, increased GRS values, regardless of whether GRS was created from donor or recipient DNA, were not associated with an increased risk of posttransplant complications (Table 4).

Analyzed individual SNPs or created GRS were not associated with ArTL, either in donors or in recipients (Table 5).

## 4. Discussion

To our knowledge, this is the first study focused on the analysis of variants within genes involved in telomere length maintenance (*TERC*, *TERF2IP*, and *OBCF1*) and their possible connection with posttransplant complications in OHT patients. Moreover, this is the only study that analyzed the genetic risk score calculated for both recipient and donor genotypes.

We identified a slight association between the donor *TERF2IP* locus (rs3784929) and AMR and between the *TERC* gene (GG homozygosity, rs12696304 polymorphism) and CGD.

AMR occurs due to a humoral immune response with antibodies binding to the endothelium on the transplanted heart [30]. Antibodies reactive against donor human leukocyte antigen molecules are termed donor-specific antibodies (DSA). DSA binding to the allograft causes myocardial injury and allograft dysfunction predominantly through immune complex activation of the classical pathway of the complement cascade and thus is a significant risk for allograft failure, CAV, and poor survival [31]. We previously reported a relationship between shorter aortic telomere length and the development of CAV in heart transplant patients [23]. Recently, a higher risk of AMR development was also observed in patients undergoing OHT following mechanical circulatory support [31]. The clinical features of graft rejection caused by de novo alloantibodies include neutrophil infiltration into capillaries in the graft, inflammation of the vascular system, fibrosis, and necrosis of blood vessel walls [32].

Oxidative stress and inflammation are major factors that accelerate biological aging, which is associated with telomere attrition [33,34]. Telomere shortening, which is induced by depletion of the shelterin complex, contributes to EC senescence. Recently, it was demonstrated that TERF2IP, a member of the shelterin complex, is an important molecular switch that simultaneously accelerates endothelial cell senescence and apoptosis [35]. The association between the donor minor allele within the *TERF2IP* locus (rs3784929) and AMR occurrence could help clarify the effect of TERF2IP on the regulation of cell senescence.

Graft dysfunction can occur early during the intraoperative period or can develop many years after transplantation. Graft dysfunction may develop as either heart failure with preserved or reduced ejection fraction, asymptomatic ventricular dysfunction, elevated intracardiac filling pressures, or depressed cardiac output on right heart catheterization [36]. Severe ischemia-reperfusion injury has been considered one of the primary mechanisms for the development of graft dysfunction. This ischemia-reperfusion injury potentiates severe inflammatory responses by upregulating various mediators leading to acute rejection as well as CAV in the long term [37]. The development of CAV lesions is preceded by endothelial dysfunction. The etiology of this allograft endothelial alteration is multifactorial and may include preexisting atherosclerosis of the graft vessels, reperfusion injury during transplantation, denervation, disruption of the lymphatic system, and acute and chronic immune injury, as well as traditional risk factors for coronary artery disease (hyperlipidemia, diabetes, hypertension, or hyperhomocysteinemia) and pathogens, such as cytomegalovirus [38].

Both rs3784929 (*TERF2IP*) and rs12696304 (*TERC*) variants are located within intron regions and thus may affect the transcription process, resulting in changes in the expression levels of the corresponding proteins, which may eventually contribute to inflammation and endothelial dysfunction during the development of AMR and CGD. It should be emphasized that we detected the relationship only in donor DNA, indicating a significant influence of the donor genome on the risk of developing posttransplant complications.

The genetic risk score analyses did not reveal a connection between the cumulative presence of risky alleles and posttransplant complications in either the donors or the recipients.

It should be noted that many previously reported studies have investigated the link between the mentioned SNPs and telomere length measured mainly in leukocytes [7,8,9,10,11,13,14,15,20,21] not in a tissue obtained as close as possible to the transplanted organ as we did in our study (aorta-heart). It is of importance, as it has been described, that there is not a strict correlation between the length of telomeres in different tissues [29,39]. Thus, the studies are not fully comparable.

Our findings could be affected only by the seemingly limited size of the study cohort, which is, in relation to the usual numbers in genetic studies, low. However, this is influenced by the fact that the number of heart transplantations is limited and that the included subjects represent more than half of the patients transplanted in the Czech Republic within the period of 10 years.

A possible association between donor *TERF2IP* (rs3784929) and AMR and between *TERC* (rs12696304) and CGD has been found. SNPs within the examined genes were not associated with aortic telomere length in transplanted patients.

## 5. Conclusions

In our robust retrospective study, which is the first focused on paired samples (recipients/donors) from OHT patients, we did not find a relationship between telomere length measured in aortic tissue and candidate SNPs within genes involved in telomere maintenance.

## Figures and Tables

**Table 1 genes-13-01855-t001:** Basic characteristics of patients.

Characteristics	Recipients	Donors	*p* Value
N (male/female)	383 (308/75)	384 (294/90)	ns
Age (years)	50.7 ± 11.9	38.8 ± 12.0	<0.0001
ArTL (T/S ratio)	0.84 ± 0.28	0.99 ± 0.31	<0.0001
Primary heart disease (N)			
Dilated cardiomyopathy	163		
Coronary artery disease	155		
Congenital heart defects	31		
Others	34		
LVAD before heart transplantation (N/%)	116 (30.3%)		
LVAD support duration (days)	201 (327.8)		
Rejection			
ACR	142		
AMR	35		
Cardiac allograft vasculopathy	58		
Chronic graft dysfunction	57		
Survival time (years)	4.7 (7.8)		
Death (N)	152		

Data are shown as the mean ± SD and factor proportion or as the median and IQR. ArTL, aortic relative telomere length; ACR, acute cellular rejection; AMR, antibody-mediated rejection; LVAD, left ventricular assist device; SD, standard deviation.

**Table 2 genes-13-01855-t002:** Frequencies of measured genotypes within candidate genes.

SNP (*Gene*)	Genotype	Recipients		Donors		*p* Value
		N	%	N	%	
**rs12696304 *(TERC)***	CC	202	53	205	53.7	0.92
	CG	155	40.7	151	39.5	
	GG	24	6.3	26	6.8	
**rs3784929 *(TERF2IP)***	AA	289	75.5	309	80.7	0.21
	AG	87	22.7	69	18	
	GG	7	1.8	5	1.3	
**rs8053257 *(TERF2IP)***	GG	337	86.7	339	88.5	0.43
	GA + AA	51	13.3	44	11.5	
**rs4387287 *(OBFC1)***	CC	232	61.4	229	60.4	0.60
	CA	136	36	135	35.6	
	AA	10	2.6	15	4	

**Table 3 genes-13-01855-t003:** Occurrence of posttransplant complications with respect to frequencies of individual genetic variants in (A) recipients and (B) donors.

(A) *Recipients*																					
SNP *(Gene)*	Genotype	ACR					AMR					CGD					CAV				
		0		1			0		1			0		1			0		1		
		N	%	N	%	*p*	N	%	N	%	*p*	N	%	N	%	*p*	N	%	N	%	*p*
**rs12696304 *(TERC)***	CC	131	64.9	71	35.1	0.38	181	89.6	21	10.4	0.78	166	82.6	35	17.4	0.43	86	72.3	33	27.7	0.10
	CG	84	57.5	62	42.5		134	91.8	12	8.2		127	87	19	13		61	71.7	24	28.2	
	GG	15	62.5	9	37.5		22	91.7	2	8.3		21	87.5	3	12.5		12	100	0	0	
**rs3784929 *(TERF2IP)***	AA	172	61.2	109	38.8	0.87	256	91.1	25	8.9	0.52	241	85.8	40	14.2	0.44	121	73.8	43	26.2	0.79
	AG	55	64	31	36		76	88.4	10	11.6		68	80	17	20		36	73.5	13	26.5	
	GG	4	57.1	3	42.9		7	100	0	0		6	85.7	1	14.3		3	60	2	40	
**rs8053257 *(TERF2IP)***	GG	198	60.9	127	39.1	0.39	296	91.1	29	8.9	0.46	274	84.6	50	15.4	0.87	136	73.1	50	26.9	0.82
	GA + AA	33	67.4	16	32.7		43	87.8	6	12.2		41	83.7	8	16.3		24	75	8	25	
**rs4387287 *(OBFC1)***	CC	142	62.8	84	37.2	0.53	206	91.2	20	8.8	0.87	185	82.2	40	17.8	0.19	100	72.5	38	27.5	0.45
	CA	82	61.2	52	38.8		120	89.6	14	10.4		117	87.3	17	12.7		55	77.5	16	22.5	
	AA	4	44.4	5	55.6		8	88.9	1	11.1		9	100	0	0		4	57.1	3	42.9	
**(B)** * **Donors** *																					
**SNP** ** *(Gene)* **	**Genotype**	**ACR**					**AMR**					**CGD**					**CAV**				
		**0**		**1**			**0**		**1**			**0**		**1**			**0**		**1**		
		**N**	**%**	**N**	**%**	** *p* **	**N**	**%**	**N**	**%**	** *p* **	**N**	**%**	**N**	**%**	** *p* **	**N**	**%**	**N**	**%**	** *p* **
**rs12696304 *(TERC)***	CC	116	58.9	81	41.1	0.44	175	88.8	22	11.2	0.34	162	81.8	36	18.2	* 0.055	81	69.8	35	30.2	0.34
	CG	98	65.3	52	34.7		140	93.3	10	6.7		131	89.7	15	10.3		65	77.4	19	22.6	
	GG	17	65.4	9	34.6		24	92.3	2	7.7		19	76	6	24		14	82.4	3	17.6	
**rs3784929 *(TERF2IP)***	AA	183	60.6	119	39.4	0.28	274	91.6	25	8.4	* 0.048	250	83.1	51	16.9	0.26	129	73.3	47	26.7	0.89
	AG	46	68.7	21	31.3		59	88.1	8	11.9		60	89.6	7	10.4		29	76.3	9	23.7	
	GG	2	40	3	60		3	60	2	40		5	100	0	0		2	66.7	1	33.3	
**rs8053257 *(TERF2IP)***	GG	204	61.3	129	38.7	0.49	303	90.9	30	9.1	0.54	279	84.1	53	15.9	0.49	143	73.3	52	26.7	0.95
	GA + AA	28	66.7	14	33.3		37	88.1	5	11.9		37	88.1	5	11.9		17	73.9	6	26.1	
**rs4387287 *(OBFC1)***	CC	139	61.8	86	38.2	0.69	207	92	18	8	0.49	193	85.8	32	14.2	0.67	103	76.3	32	23.7	0.11
	CA	84	64.1	47	35.9		116	88.6	15	11.4		107	82.3	23	17.7		51	69.9	22	30.1	
	AA	8	53.3	7	46.7		13	86.7	2	13.3		13	86.7	2	13.3		3	42.9	4	57.1	

ACR, acute cellular rejection; AMR, antibody-mediated rejection; CAV’ cardiac allograft vasculopathy; CGD, chronic graft dysfunction; SNP, single nucleotide polymorphism; 0—without rejection; 1—rejection occurrence. * Indicates significance at *p* < 0.05.

**Table 4 genes-13-01855-t004:** Association between GRS and posttransplant events. The GRS, genetic risk score, expressed the sum of the risk alleles of all measured SNPs.

GRS *Recipients*	ACR					AMR					CGD					CAV				
	0		1			0		1			0		1			0		1		
	N	%	N	%	*p*	N	%	N	%	*p*	N	%	N	%	*p*	N	%	N	%	*p*
0	62	70.5	26	29.5	0.79	81	92.1	7	7.9	0.64	71	80.7	17	19.3	0.43	40	69	18	31	0.53
1	82	57.3	61	42.7		131	91.6	12	8.4		118	83.1	24	16.9		57	75	19	25	
2	40	55.6	32	44.4		64	88.9	8	11.1		70	97.2	2	2.8		30	76.9	9	23.1	
3	21	70	9	30		26	86.7	4	13.3		20	66.7	10	33.3		13	61.9	8	38.1	
4	15	62.5	9	37.5		22	91.7	2	8.3		22	91.7	2	8.3		12	85.7	2	14.3	
5	8	72.7	3	27.3		10	90.9	1	9.1		9	81.8	2	18.2		4	66.7	2	33.3	
6	1	50	1	50		2	100	0	0		2	100	0	0		2	100	0	0	
**GRS** ** *Donors* **	**ACR**					**AMR**					**CGD**					**CAV**				
	**0**		**1**			**0**		**1**			**0**		**1**			**0**		**1**		
	**N**	**%**	**N**	**%**	** *p* **	**N**	**%**	**N**	**%**	** *p* **	**N**	**%**	**N**	**%**	** *p* **	**N**	**%**	**N**	**%**	** *p* **
0	55	59.8	37	40.2	0.20	80	87	12	13	0.38	76	82.6	16	17.4	0.29	39	69.6	17	30.4	0.93
1	78	55.3	63	44.7		132	93.6	9	6.4		121	85.2	21	14.8		66	77.7	19	22.3	
2	58	71.6	23	28.4		75	92.6	6	7.4		67	84.8	12	15.2		28	68.3	13	21.7	
3	21	75	7	25		26	92.9	2	7.1		21	75	7	25		13	68.4	6	31.6	
4	11	61.1	7	38.9		16	88.9	2	11.1		17	94.4	1	5.6		6	100	0	0	
5	7	58.3	5	41.7		8	66.7	4	33.3		12	100	0	0		6	66.7	3	33.3	

**Table 5 genes-13-01855-t005:** Distributions of telomere length according to (A) individual measured genotypes and (B) GRS. GRS expressed the sum of risky alleles of all measured SNPs. ArTL is given as the mean ± SD or mean ± SE.

SNPs (*Gene*)	Genotype	(A) *Recipients*			*Donors*		
		N	Mean ± SD	*p*	N	Mean ± SD	*p*
**rs12696304 *(TERC)***	CC	197	0.84 ± 0.28	0.95	194	0.98 ± 0.30	0.41
	CG	149	0.84 ± 0.30		143	0.99 ± 0.32	
	GG	23	0.82 ± 0.21		23	1.09 ± 0.39	
**rs3784929 *(TERF2IP)***	GG + AG	92	0.84 ± 0.32	0.75	70	0.99 ± 0.32	0.78
	AA	279	0.84 ± 0.27		290	0.99 ± 0.32	
**rs8053257 *(TERF2IP)***	AA + AG	50	0.83 ± 0.27	0.80	41	0.91 ± 0.30	0.06
	GG	321	0.84 ± 0.29		320	1.01 ± 0.32	
**rs4387287 *(OBFC1)***	AA + AC	143	0.81 ± 0.26	0.12	144	0.97 ± 0.31	0.17
	CC	224	0.86 ± 0.29		212	1.02 ± 0.31	
	CC	224	0.86 ± 0.29		212	1.02 ± 0.31	
**GRS**	**(B) *Recipients***				** *Donors* **		
	**N**		**Mean ± SE**	** *p* **	**N**	**Mean ± SE**	** *p* **
**0**	86		0.89 ± 0.03	ns	90	1.03 ± 0.03	ns
**1**	145		0.85 ± 0.03		133	0.93 ± 0.03	
**2**	72		0.83 ± 0.04		82	1.06 ± 0.04	
**3+**	68		0.85 ± 0.04		55	0.95 ± 0.04	

## Data Availability

All data that support the findings of this study are available from the corresponding author upon reasonable request.

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
