# Peer review of "Posttransplant Complications and Genetic Loci Involved in Telomere Maintenance in Heart Transplant Patients"

_genes, 2022, doi:10.3390/genes13101855_

Round 1

Reviewer 1 Report

Review of article

„Posttransplant complications and genetic loci involved in telomere maintenance in heart transplant patients”  by Dana Dlouha et al.

The present study is a first work focused on paired samples (recipients/donors) from OHT (orthotopic heart transplantation) patients.

Authors did not find a relationship between telomere length measured in aortic  tissue (ArTL (T/S ratio)) and candidate SNPs within genes involved in telomere maintenance. Polymorphisms in the donor genes were associated with AMR and CGD. Other polymorphisms within the recipient's genes were not statistically significant.

Major revision

- In lines 91-98 the authors state that they isolated DNA by a modified proteinase K method. Was the isolated DNA not degraded? DNA degradation can affect telomere length determination.

- What are the causes of the variation in the number of patients in Table 2 and Table 3? Example: Table 2, the number of rs12696304 (TERC) CG - 155 recipients. Table 3, the number of rs12696304 (TERC) CG - 146 recipients?

Author Response

Dear reviewer, thank you for your comment.

As we reported genomic DNA was isolated from a maximum of 100 mg of the aortic samples using the modified “salt out” method [Miller at el. 1988] after cell lysis by Proteinase K. Proteinase K is a broad-range endolytic protease widely used for digestion of proteins in nucleic acid preparations. It degrades proteins (and only proteins) even in the presence of detergents. Proteinase K cleaves peptide bonds at the carboxylic sides of aliphatic, aromatic, or hydrophobic amino acids. So, DNA is not affect using proteases. Moreover, the quantity and purity of isolated DNA were examined by standard spectrophotometry and DNA integrity was tested on a 0.7% agarose gel (lines 97-99). All degraded samples or samples with low quality were excluded from analysis as we reported in lines 80-81.

- What are the causes of the variation in the number of patients in Table 2 and Table 3? Example: Table 2, the number of rs12696304 (TERC) CG - 155 recipients. Table 3, the number of rs12696304 (TERC) CG - 146 recipients?

The differences in the number of individuals are due to the fact, that some patients were not followed-up in our institute after Tx (e. g. the foreigners). Therefore, we lack information on the occurrence of post-transplantation complications in these patients.

Reviewer 2 Report

Congratulations on the study you demonstrate a slight association between the donor TERF2IP locus (rs3784929) and AMR and between the TERC gene (GG homozygosity, rs12696304 polymorphism) and CGD. Demonstrating the association of variants within genes involved in telomere length maintenance (TERC, TERF2IP, and OBCF1) and their possible connection with posttransplant complications in OHT patients is an important achievement. The association between the donor minor allele within the TERF2IP locus (rs3784929) and AMR occurrence could help clarify the effect of TERF2IP on the regulation of cell senescence. An important finding is that you detected the relationship between  rs3784929 (TERF2IP) and rs12696304 (TERC) with the development of inflammation and endothelial dysfunction during the development of AMR and CGD only in donor DNA, indicating a significant influence of the donor genome on the risk of developing posttransplant complications. A possible association between donor TERF2IP (rs3784929) and AMR and between TERC (rs12696304) and CGD has been found and now you have to answer the limitations of the study:  the limited size of the study cohort and the existence of discrepancies in correlation linkage in the length of telomeres in different tissues.

Author Response

Dear reviewer, thank you for your comment.

The limited size of the study cohort is relative. It is one of the largest cohorts of heart transplant patients, where also genotypes from donors are available. Our study is a single center and includes approximately 380 individuals in each recipient vs. donor group collected during one decade and represents the majority of subjects transplanted within the entire Czech Republic.

The relative length of telomeres measured in peripheral blood leukocytes is a commonly used system marker for biological aging and can also be used as a biomarker of cardiovascular aging. We previously measured relative telomere length (rTL) in twelve different human tissues (peripheral blood leukocytes, liver, kidney, heart, spleen, brain, skin, triceps, tongue mucosa, intercostal skeletal muscle, subcutaneous fat, and abdominal fat) from twelve subjects (Dlouha et al. 2014). This study demonstrated the highest rTL variability in peripheral leukocytes and the lowest variability in the brain. We found a significant linear correlation only between leukocyte rTL and both intercostal muscle and liver rTL. Further, in a cohort of heart Tx patients (N=73), we compared rTL and aortic telomere length and found a significant inverse correlation (Dlouha et al. 2016). Another study by Yin H et al. 2018 reported a positive correlation between leukocyte telomere length and skeletal muscle TL and right atrium TL in patients after cardiac surgery. The text was slightly modified (lines 231-243).

Dlouha D, Maluskova J, Kralova Lesna I, Lanska V, Hubacek JA. Comparison of the relative telomere length measured in leukocytes and eleven different human tissues. Physiol Res. 2014;63(Suppl 3):S343-50. doi: 10.33549/physiolres.932856.

Dlouha D, Vymetalova J, Malek I, Hubacek JA. Comparison of relative telomere length measured in aortic tissue and leukocytes in patients with end stage heart failure. Neuro Endocrinol Lett. 2016;37(2):124-8. PMID: 27179575.

Yin, H.; Akawi, O.; Fox, S.A.; Li, F.; O'Neil, C.; Balint, B.; Arpino, J.M.; Watson, A.; Wong, J.; Guo, L.; Quantz, M.A.; Nagpal, A.D.; Kiaii, B.; Chu, M.W.A.; Pickering, J.G. Cardiac-Referenced Leukocyte Telomere Length and Outcomes After Cardiovascular Surgery. JACC Basic Transl Sci 2018, 3, 591-600, doi: 10.1016/j.jacbts.2018.07.004